# Predicting the Burden for Surgical Aortic Valve Replacement in a Tertiary Centre: The Impact of Aged Populations for the Next Decades

**DOI:** 10.3390/jcm14103365

**Published:** 2025-05-12

**Authors:** Rafael Maniés Pereira, Nuno Guerra, João Moreira Gonçalves, Ricardo Ferreira, Ângelo Nobre, Dulce Brito, Teresa Ferreira Rodrigues, Tiago R. Velho

**Affiliations:** 1Cardiothoracic Surgery Research Unit, Centro Cardiovascular da Universidade de Lisboa (CCUL@RISE), Faculdade de Medicina da Universidade de Lisboa, 1649-028 Lisbon, Portugal; rafaelmanies@edu.ulisboa.pt; 2Escola Superior de Saúde da Cruz Vermelha, 1350-125 Lisbon, Portugal; 3Cardiothoracic Surgery Department, Hospital de Santa Maria, 1649-035 Lisbon, Portugal; nmncguerra@gmail.com (N.G.); joao.dmoreiragoncalves@gmail.com (J.M.G.); rmferreirast@gmail.com (R.F.); angelolucasnobre@gmail.com (Â.N.); 4Cardiology Department, Hospital de Santa Maria, 1649-035 Lisbon, Portugal; dulcebrito59@gmail.com; 5Faculdade de Ciências Sociais e Humanas, Universidade Nova de Lisboa, 1099-085 Lisbon, Portugal; trodrigues@fcsh.unl.pt

**Keywords:** aortic stenosis, population aging, aging index, longevity index, healthcare organization

## Abstract

**Background/Objectives:** The incidence of aortic stenosis (AS) is predicted to rise with the aging population, emerging as a growing public health challenge in developed countries, leading to an increased demand for intervention. Our aim is to predict the evolution of proposed cases for SAVR in the geographic referral area of our tertiary hospital until 2041. **Methods:** We used data from the Portuguese Census for 2001, 2011, and 2021 to analyze the resident population within the Cardiothoracic Surgery Department’s referral area. Applying population projection methods (rate of geometric growth), we projected demographic trends over 20 years, from 2021 to 2041. Our analysis focused on AS cases who underwent SAVR in our department between 2001–2011 and 2011–2021. **Results:** Between 2001 and 2021, there was an increase in the overall population, particularly among the elderly (1.4% growth rate in the population ≥ 65 years old). The aging index increased from 128.4 (110.5–180.6) in 2001 to 189.1 (155.9–222.5) in 2021 (*p*-value < 0.001). Similarly, the longevity index significantly increased between 2001 [42.6 (40.8–44.80)] and 2021 [49.30 (47.7–51.8)] (*p*-value < 0.001). The number of SAVRs performed increased, with a mean increase of 8.11 surgeries/year (R^2^ = 0.6457, *p* < 0.001). By 2041, our referral center will increase SAVR by at least 51 surgeries/year in a decreasing growth rate scenario, and 67 surgeries/year in a growth rate stagnation scenario. **Conclusions:** The ongoing trend of population aging will increase the demand for healthcare resources, particularly within the cardiovascular domain. Accurately assessing the volume of SAVR is imperative for reformulating strategies to address the increasing demand effectively.

## 1. Introduction

Aortic stenosis (AS) is the most common heart valve disease in Europe and North America [1,2,3], and it is becoming a significant public health concern in developed countries [4]. This is primarily due to the shift from rheumatic to degenerative valve disease, which is closely linked to the aging population. Predictions suggest that the incidence of AS will increase due to this demographic shift [4,5].

Epidemiological studies have shown that at least one in every eight people over 75 years old has moderate to severe aortic stenosis [6,7]. This progressive disease has high mortality rates if left untreated [4,6]. When symptoms appear, the average survival rate is about 50% at two years and 20% at five years [8,9]. Currently, there are no drugs capable of slowing the progression of AS, so valve replacement is the only targeted therapeutic option available [10]. Advancements in cardiac surgery, reductions in morbidity and mortality, and improvement in patient selection have lessened age as a limiting factor for surgery, resulting in an increase in interventions for aortic stenosis in surgical departments [11].

Regardless of the approach used (surgical aortic valve replacement—SAVR, or transcatheter aortic valve implantation—TAVI), treatment for AS is not always immediate and is often associated with waiting times and waiting lists. Acknowledging its high mortality rate, a Working Group on Waiting Times for Cardiac Surgery, jointly appointed by the Portuguese Society of Cardiothoracic and Vascular Surgery (SPCCTV) and the Portuguese Society of Cardiology (SPC) established in 2015, reported mortality rates of 3.7% at one month and 8.0% at six months for individuals on the waiting list [9,12]. Additionally, there was an increase in operative mortality among patients with severe and symptomatic aortic stenosis who experienced deteriorated left ventricular function due to waiting time and/or delayed referral [12].

This study aims to predict the evolution of proposed cases for SAVR in the geographical referral area of a tertiary hospital in Lisbon, Portugal. The goal is to comprehensively characterize the social and demographic aspects of the study population and project trends up to 2041. Recognizing the impact of waiting times, associating demographic studies with a prospective analysis is crucial for assessing the disease’s impact on potentially affected healthcare facilities, enabling us to predict and plan for future needs more effectively.

## 2. Materials and Methods

The dataset consists of patients diagnosed with severe AS who were recommended for SAVR within the geographical referral area of the Cardiothoracic Surgery Department at Centro Hospitalar e Universitário de Lisboa Norte—Hospital de Santa Maria, E.P.E., Lisbon, Portugal. The referral network, organized by geographical distribution, is available on the National Health Service Portal, Hospital Referral Networks [11], and received approval on 19 December 2017.

We summarized the entire population within the referral area of the Cardiothoracic Surgery Department of the hospital under study in Appendix A. The table was created by consolidating information from various institutional pages of each referring hospital, covering all the municipalities served by each of the hospitals.

This study received approval from the local Ethics Committee (Comissão Ética Centro Hospitalar Lisboa Norte—Ref. No. 23/18) and followed the Strengthening the Reporting of Observational Studies in Epidemiology guidelines. Because of the nature of our study, written informed consent was waived.

The collection of data regarding the resident population in each of the municipalities was carried out using information from the census, available at the online platform PORDATA©—Contemporary Portugal Database. Periods from the last three population censuses were selected, namely for the years 2001, 2011, and 2021. The information collected from the censuses represents current and recent past demographic data, allowing for a retrospective time frame of 20 years. Calculation of aging and longevity indexes, geometric growth rate, doubling time in years, projections on population evolution, and estimation of interventions due to aortic stenosis was performed with data provided by PORDATA©. The aging index is a demographic indicator that compares the elderly population to the youth population. It is calculated by dividing the number of individuals aged 65 and older by the number of individuals aged 0 to 14 and then multiplying the result by 100. An aging index greater than 100 indicates that there are more elderly individuals than young individuals in the population. This index is used to assess the relative aging of a population, with higher values signifying a greater proportion of elderly people [13]. The longevity index measures the proportion of the elderly population (aged 65 and older) that is “very old” (aged 75 and older). It is calculated by dividing the number of individuals aged 75 and older by the number of individuals aged 65 and older. A higher value for the longevity index reflects a larger proportion of elderly individuals who are 75 years or older, indicating an aging population with a greater share of individuals in advanced old age [13].

In terms of geography, we conducted a search on the platform by municipality, covering all 41 municipalities outlined in Appendix A. Instead of utilizing the Nomenclature of Territorial Units for Statistical Purposes (NUTS) levels, such as NUTS I, II, or III, we opted to select data by municipality to align with the specific scope of our study.

Demographic data regarding patients were retrieved from the database (CardioBase^®^) of the Cardiothoracic Surgery Department, Hospital de Santa Maria, Lisbon, Portugal. The search was refined to include patients with aortic stenosis (isolated or combined) who underwent SAVR during the two decades under study, 2001–2011 and 2011–2021. To enhance the association of the disease with population aging, mitigating the bias of other structural cardiac changes, we only included patients aged 65 or older at the time of surgery. Surgeries such as reoperation, emerging surgeries, SAVR by endocarditis, and patients with only aortic insufficiency were excluded.

### Statistical Analysis

We used mathematical population projection methods to achieve our goal of projecting population changes over a 20-year period, from 2021 to 2041. We employed a mathematical formula to determine the rate of geometric growth over the decades we analyzed. To provide more accurate population estimates, we used two different scenarios in our calculations: (i) a continued decrease in the growth rate and (ii) a stagnation in the growth rate. These formulas were applied from 2001 to 2011 and then from 2011 to 2021 to assess the pace of growth. Our study focused on calculating indicators for the entire population and specifically on the elderly. We also calculated the doubling time in years using mathematical methods.

To compare aging and longevity indices across the studied years, we conducted the Kruskal−Wallis test. When we identified significant differences in the Kruskal−Wallis test, we performed ad hoc comparisons using Dunn’s multiple median comparison test, adjusted with the Bonferroni method to control type I error. The analyses were carried out using the R programming environment (R Studio 4.3.1) [14]. We utilized the following R packages: *ggplot2* [15], *dplyr* [16], *tidyr* [17], and *paleteer* [18]. Our chosen significance level for the study was 5%.

## 3. Results

### 3.1. Population

The changes in the population being studied are outlined in Appendix A, which contains data from the census [19]. This table provides information about the local resident population, including breakdowns by age groups (0–14 years, 15–64 years, and 65 years and older). It is worth noting that both the total population and the population of individuals aged 65 and older have consistently increased in recent years. According to the 2021 census, the population aged 65 and older has exceeded half a million people (535,894) in our department’s geographical area. In contrast, the population aged 0–14 years and 15–64 years saw an increase between 2001 and 2011, but declined in subsequent censuses.

Based on this information, we created Figure 1 for the 41 municipalities studied, focusing on individuals aged 65 and older using 2021 census data. This figure highlights the top five municipalities with the highest number of elderly individuals in our center, which are Lisboa, Loures, Almada, Seixal, and Odivelas.

#### 3.1.1. Population by Age Groups and Gender

To illustrate changes in the population over time, we analyzed data from the 2001 to 2021 censuses, as detailed in Appendix A. During this time frame, both the male (1,042,214 vs. 1,103,886) and female (1,125,005 vs. 1,215,430) populations experienced an overall increase [20]. Across different age groups, growth was observed in both genders, except for individuals aged 15–34 years. We created population pyramid charts for the years 2001, 2011, and 2021 (Figure 2) to represent information based on age and gender. All of the pyramids showed an urn shape, indicating a population with a smaller number of young people and a larger proportion of elderly individuals. From a demographic perspective, this pyramid shape suggests a population with low birth rates and low mortality, features commonly found in developed countries. Based on the demographic transition phases, it can be concluded that the studied population is in a prolonged and stable life cycle, characterized by a scarcity of young individuals and an increasing elderly population. This pattern aligns with characteristics observed in the later stages of the demographic transition, similar to phases 4 and 5, where both birth and death rates are low. This results in a stabilization of the population with only modest growth, typical of most developed countries [21].

#### 3.1.2. Aging Index

The aging index, a crucial metric for measuring aging, was analyzed for all 41 municipalities in the study [13]. Since 2001, the aging index of the population has steadily and significantly increased, as shown in Figure 3 (*** *p*-value < 0.001). Specifically, a comparison of median values between 2011 (128.4) and 2021 (189.1) revealed a substantial increase (*** *p*-value < 0.001).

There are significant differences in aging among the municipalities. In 2021, the lowest aging index was observed in Mafra (108.5), while the highest was in Alcoutim (758.9). This implies that for every 100 young individuals in Alcoutim, there were approximately 758.9 elderly individuals. In 2001, only 9 out of the 41 studied municipalities had an aging index below 100 (Alcochete, Loures, Mafra, Moita, Odivelas, Seixal, Sines, Albufeira, and Lagoa), indicating more elderly than young individuals in the remaining 32 municipalities. By 2011, the number of municipalities with an aging index below 100 decreased to 4 (Alcochete, Mafra, Seixal, and Albufeira). According to the 2021 data, none of the studied municipalities will have an aging index below 100, highlighting the evident aging trend in the population under study.

#### 3.1.3. Longevity Index

The longevity index was analyzed for all of the municipalities studied [11], and showed a statistically significant increase between 2001 and 2021 (*p*-value < 0.001). The increase was particularly notable between 2001 (median 42.6) and 2011 (median 49.6) (*p* < 0.001). In both 2001 and 2011, Seixal had the lowest longevity index, with values of 35.6 and 39.7, respectively. On the other hand, in 2001, São Brás de Alportel had the highest longevity index (47.7). This indicated that approximately half of the elderly population consisted of younger elderly (≥65 years old), while the other half consisted of older elderly (≥75 years old). The evolution of this index is depicted in Figure 3.

### 3.2. Prospective Analysis Applied to the Studied Population

#### 3.2.1. Geometric Growth Rate

Between 2001 and 2011, it was noted that the total population under study increased by 0.49% per year per 100 people, but this growth rate slowed down to 0.19% between 2011 and 2021. However, the decline in growth rate for the elderly population is more noticeable than that observed for the general population, with a decrease from 1.69% for the period 2001–2011 to 1.40% for 2011–2021.

#### 3.2.2. Doubling Time in Years

The population aged 65 and over is expected to double from the observed growth rates during the 2011–2021 decade, assuming constancy. With a growth rate of 1.4%, it is projected that this population will reach over one million individuals by 2071, with an estimated count of 1,066,202. This doubling is expected to take approximately 50 years.

#### 3.2.3. Estimation of Population Evolution

We used mathematical methods to project the future population in the area served by our hospital. This projection covers a 20-year period and includes scenarios for the year 2041. We specifically focused on the association between aortic stenosis and the aging demographic and calculated the impact on the entire population as well as individuals aged 65 and over.

To make the estimation more realistic and adaptable, we applied two different scenarios to the calculations: a continued decrease in the growth rate and a scenario where the growth rate stagnates. Instead of directly estimating the population based on the 2021 census, we used a two-phase estimation method. First, we covered the period from 2021 to 2031, and then applied the growth rate reduction for the subsequent phase from 2031 to 2041. We chose this method to be conservative in our calculations, aiming to avoid overestimating the population growth.

#### 3.2.4. Scenario of Decreasing Growth Rate

In the scenario of a continued decrease in growth rate, it was assumed that the reduction in the growth rate for the two upcoming decades (2021–2031 and 2031–2041) would occur in the same proportion as the reduction observed between the decades of 2001–2011 and 2011–2021. Consequently, the results presented reflected a reduction in the growth rate of 0.3% for the total population and 0.29% for the population aged ≥65 years old. The estimated values for the number of inhabitants for the population and for each of the decades are described in Table 1.

#### 3.2.5. Scenario of Growth Rate Stagnation

In a situation of stagnant growth rates, it was assumed that the growth rate would remain constant at the same rate observed in the 2011–2021 decade, specifically 0.19% for the entire population and 1.40% for the population aged 65 years and older. The estimated values for the population and for each of the decades are detailed in Table 1.

#### 3.2.6. Estimation of the Number of Interventions in 2041

Between 2001 and 2011, a total of 1538 patients aged 65 and over with aortic stenosis (AS) underwent surgical intervention. This number increased to 2424 patients with similar characteristics who underwent surgical intervention between 2011 and 2021.

To achieve the study objective, the prevalence of aortic stenosis cases in patients aged 65 and over who underwent cardiac surgery was calculated for the two decades. A prevalence of 0.33% was observed from 2001 to 2011, which increased to 0.45% from 2011 to 2021. To estimate the number of surgeries required until 2041, the last recorded prevalence (0.45%) was applied to each of the estimated populations aged 65 and over that were calculated earlier (Table 2). If a scenario of decreasing growth rate is confirmed and the elderly population increases to 650,506, an increase of 503 surgical aortic valve replacements (SAVR) relative to the 2011–2021 period is anticipated for each decade. This implies that, on average, the referral center will need to increase its rate of SAVR by at least 51 surgeries per year. On the other hand, if a scenario of growth rate stagnation is confirmed and the elderly population increases to 688,335, an increase of 674 SAVR relative to the 2011–2021 period is anticipated for each decade. This suggests that, on average, the referral center will need to increase its rate of SAVR by at least 67 surgeries per year.

## 4. Discussion

The results of our study indicate that the aging population will significantly increase the need for SAVR at our tertiary center in the coming decades. This emphasizes the necessity for well-informed policy-making and strategic planning to effectively manage the anticipated rise in surgical procedures. We have observed a consistent increase in the resident population within our center’s geographical area over the past 20 years. Notably, this increase is primarily due to a higher growth rate in the elderly population compared to the overall population. By analyzing this data alongside the rise in longevity and aging indices, it can be concluded that the population growth is partly a result of an expansion in the older age groups, characterized by both an increase in the number of elderly individuals and an upward shift in their age demographic.

Population aging is a global demographic trend characterized by both an aging base and a growing elderly population. Factors such as declining fertility rates, improved healthcare, and increased life expectancy contribute to a gradual decline in the young population [22], coupled with an increase in the elderly population [23,24]. This demographic shift not only has societal implications, but also introduces new challenges for healthcare services [25]. Additionally, it is important to consider the possibility of paradoxical situations in certain regions, where the aging index increases without a corresponding rise in longevity index. This can lead to a scenario where the elderly population increases in number but not in age. In high-income countries, the average age of first onset and death from cardiovascular diseases has been increasing, shifting the burden of these diseases to older age groups [16]. In fact, 23.1% of the total disease burden is attributable to disorders in people aged 60 years and older, with cardiovascular diseases being a significant contributor [26]. This burden leads to a substantial financial impact; in the United Kingdom, healthcare-associated costs are estimated to annually rise between 0.48% to 1.12% due to the aging population [27]. These trends underscore the importance of proactive planning and resource allocation to address the evolving healthcare needs associated with an aging population.

The importance of health is evident in all societies, regardless of their level of development [24]. However, the utilization and access to the healthcare system vary across countries or regions, emphasizing the necessity of conducting multiple estimation studies in different populations. Portugal, in particular, exhibits distinct patterns in healthcare utilization. It stands out as one of the countries with intensive use of healthcare services by individuals aged 60–84 years [28]. Moreover, the healthy life expectancy at 65 years old in Portugal, representing the number of years an individual is expected to live without significant health-related physical limitations after turning 65 years old, is approximately 6.9 years on average. This figure is notably below the average of the Organization for Economic Co-operation and Development (OECD) and highly developed countries like Norway. The lower healthy life expectancy in Portugal is directly linked to an increased demand for access to the healthcare system and a subsequent rise in healthcare-related costs [23]. Understanding these nuances in healthcare dynamics is crucial for effective healthcare planning and resource allocation [29].

The increasing demand for healthcare in cardiovascular medicine is a major concern in Portugal, as emphasized in a recent position document by the National Network of Hospital Specialties and Referrals [11]. The document acknowledges that both coronary and valvular heart diseases are expected to increase in prevalence in Portugal due to population aging. Therefore, it is crucial to thoroughly examine and discuss how this heightened demand will be seen in specific populations. This detailed understanding is essential for formulating appropriate strategies, including the allocation of human and material resources, to effectively address the evolving healthcare needs.

At the broader European level, the population is aging very quickly, and the projections are that by 2050, the number of people aged 60 and over living in the WHO European Region will exceed 300 million [30]. In addition, the disease burden of aging is on the rise, most notably from cardiovascular disease, as in one such study between 1990 and 2019, where the rising burden of care in health is shown, particularly in Eastern Europe where there is regional variation. This indicates the necessity to invest in resources for health promotions and treatment, notably to lessen the burden of years of life lost (YLL) and years lived with disability (YLD) [31]. On top of this, advances in technology such as eHealth and assistive technologies are increasingly important in the provision of care for the elderly and the management of long-term disease with the aim to encourage healthier aging [32].

### Limitations

It is important to note that the Hospital de Loures, PPP is referred to by two municipalities (Mafra and Loures), but these do not cover the entire area of the municipalities. As it is not possible to gather demographic population information at this geographical level, the decision was made to include the population of the entire municipality.

There are other limitations to consider in this section. For instance, there is a possibility that some patients referred out of the geographical area might seek treatment in the private health sector or request referral to another hospital. Another important factor to consider is the period of this study, which covers the period from 2001 to 2041. This long period is characterized by significant advancements in technology for diagnosis and access to healthcare that may influence changes in a better capacity to diagnose aortic stenosis. Additionally, it is possible that some patients may transition to private healthcare as the demand for SAVR increases, potentially affecting the volume of procedures in the public sector. Finally, the expanding role of transcatheter aortic valve implantation (TAVI) may influence future SAVR demand. As TAVI eligibility grows, particularly for older and higher-risk patients, some individuals previously treated with SAVR may now be candidates for TAVI. This shift could reduce the number of SAVR procedures required, and further research is needed to assess TAVI’s impact on long-term healthcare planning and resource allocation for aortic valve replacements.

## 5. Conclusions

Population aging represents a prevailing demographic trend that will induce substantial chances in access to and demand for healthcare resources, especially in the cardiovascular field. Our tertiary center foresees an estimated increase in the demand for SAVR by 2041, contingent on the population growth rate. The evaluation of SAVR volume in specific populations is pivotal for formulating appropriate strategies to effectively respond to these evolving demands.

## Figures and Tables

**Figure 1 jcm-14-03365-f001:**
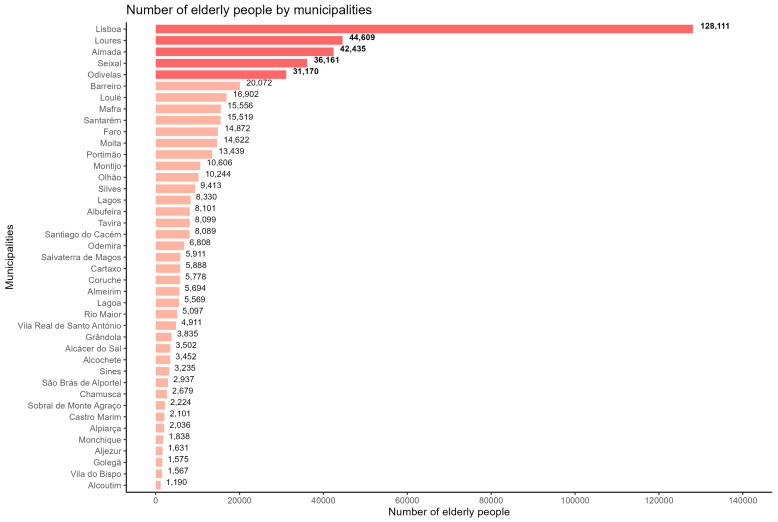
Number of elderly by municipalities in 2021 in the geographical reference area (source: census).

**Figure 2 jcm-14-03365-f002:**
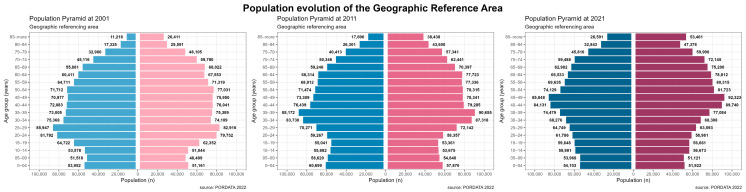
Population evolution of the geographical reference area from 2001 to 2021.

**Figure 3 jcm-14-03365-f003:**
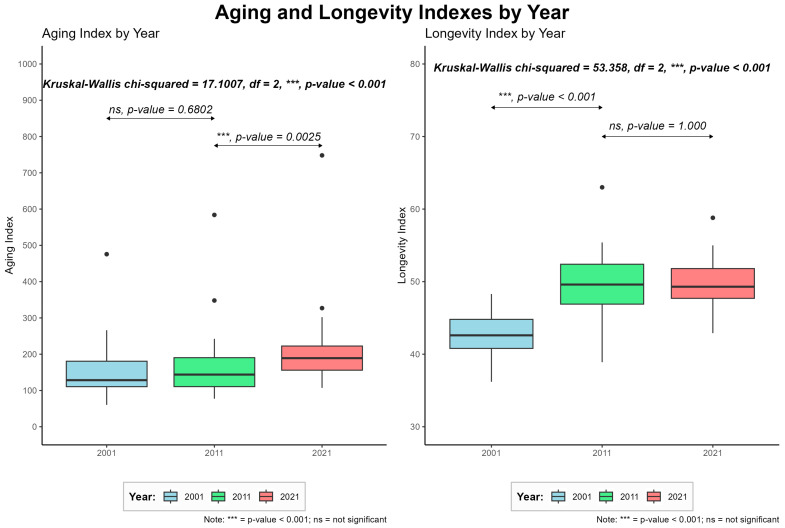
Evolution of aging and longevity indexes by years.

**Table 1 jcm-14-03365-t001:** Estimated values for the population in a decreasing growth rate and in a growth rate stagnation scenario.

		2011–2021	2021–2031	2031–2041
Information of census	Total population	2,319,319	-	-
	Population ≥ 65 years	535,894	-	-
Decreasing growth rate (estimation)	Total population	-	2,295,039	2,204,468
	Population ≥ 65 years	-	598,840	650,506
Growth rate stagnation (estimation)	Total population	-	2,364,114	2,339,368
	Population ≥ 65 years	-	615,982	688,335

**Table 2 jcm-14-03365-t002:** Projection of the number of interventions in 2041.

	Estimated Population	Estimated Number of Surgeries in 2031–2041	Estimate of Additional Surgeries per Year in This Decade
**Decreasing growth rate scenario**	650,506	2927	+51
**Growth rate stagnation scenario**	688,335	3098	+67

## Data Availability

The data supporting the findings of this study are available from the corresponding author upon reasonable request.

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
