# Peer review of "Predicting the Burden for Surgical Aortic Valve Replacement in a Tertiary Centre: The Impact of Aged Populations for the Next Decades"

_jcm, 2025, doi:10.3390/jcm14103365_

Round 1
Reviewer 1 Report
Comments and Suggestions for Authors
Interesting article
Explain better what ageing index and longevity index are
Robust analysis based on three decades
Are there countries where aging index increases without a significant increase in longevity index? Are there paradoxical situations?
Explain well that they are derived from the cut-offs of 65 years and 85 years
To increase the generazability of the study, can a formula be derived to predict the increased need for aortic valve replacements?
No reference to the TAVI world. Don't you think that some of these SAVRs actually become TAVRs?
I would try to make less local and broader reasoning
A brief discussion on European needs (and not only referring to your hospital)
Can we foresee a transition of these patients to private healthcare?
Author Response
|
Comments 1: Interesting article |
|
Response 1: Thank you for the positive comment. We are glad that you found the article interesting.
|
|
Comments 2: Explain better what ageing index and longevity index are |
|
Response 2: Thank you for the suggestion. We've revamped the manuscript to include an extended explanation of both indices. The ageing index is the ratio of the number of elderly people (65 and above) to the number of young people (0-14), multiplied by 100. A number greater than 100 shows the presence of more elderly people than young people. The longevity index, however, is the proportion of the elderly population (65 and above) that is very old (75 and above). It is calculated by taking the number of people aged 75 and above and dividing it by the number of people aged 65 and above. We've demystified these definitions in the Methods section to enhance knowledge. These modifications have been highlighted in yellow in the manuscript for your reference (lines 98-109).
|
|
Comments 3: Robust analysis based on three decades |
|
Response 3: Thank you for the comment. We are glad you acknowledged the consistency of our analysis, which is spread across three decades of population and surgical data. The long-term view allows us to get at meaningful trends in population aging and its effect on the demand for surgical aortic valve replacements (SAVR), giving us an overall picture of the changing healthcare demand. We feel that the large timeframe fortifies the integrity of our forecasts for future decades.
|
|
Comments 4: Are there countries where aging index increases without a significant increase in longevity index? Are there paradoxical situations? |
|
Response 4: I appreciate this question. In response to it, the Discussion section has now covered the possibility of occurrence of paradoxical circumstances where the aging index rises without an increase in the longevity index. While an increase in the aging index and longevity index occurs in most developed nations through better health care and an increase in longevity, there are instances where the aging index is higher because of an increase in the number of elderly individuals but without an increase in longevity. It happens in nations where there is unequal provision of health care, the life expectancy is not very high, or where the elderly group shows higher mortality rates, and it can lead to the situation where the population of the elderly is on the rise but not the age. We now include a short discussion of such paradoxical circumstances, recognizing the intricacy of demographical trends in different regions of the globe. These modifications have been highlighted in yellow in the manuscript for your reference (lines 299-302).
|
|
Comments 5: Explain well that they are derived from the cut-offs of 65 years and 85 years |
|
Response 5: Thank you for this useful comment. We have updated the manuscript to include a clearer description of how the aging index and longevity index are calculated from the specific cut-offs of 65 years and 85 years. The aging index is determined by taking the population aged 65 and older, and dividing it by the population aged 0-14 years, and then multiplying the result by 100, to reflect the proportion of elderly compared to youth. The longevity index is the proportion of the population aged 65 and older who are 85 and older. It is determined by taking the number of the population aged 85 and older and dividing by the number of the population aged 65 and older. These cut-points are essential in order to define the distinction between overall aging (65 years and greater) and the portion of the population that has attained very old age (85 years and greater). We have clarified these definitions and how they are determined in the Methods section of the manuscript to provide complete understanding. These changes have been highlighted in yellow in the manuscript for your reference and were changed together with those in comment 2 (lines 98-109).
|
|
Comments 6: To increase the generazability of the study, can a formula be derived to predict the increased need for aortic valve replacements? |
|
Response 6: We appreciate the significance of this proposal. To enhance the generalizability of our work, we wondered whether it would be relevant to incorporate a figure showing the formula employed to project the population of the elderly. However, we decided not to include this figure in the current version of the manuscript. We believe that such a visual representation could provide better insight into how we are projecting future demand for SAVR based on population trends. While we did not include the formula at this stage, we understand that it may offer a clearer understanding of the calculations involved and how these projections can be used for future healthcare planning. We would be happy to add this figure in a future revision or in supplementary materials if deemed necessary.
|
|
Comments 7: No reference to the TAVI world. Don't you think that some of these SAVRs actually become TAVRs? |
|
Response 7: Thank you for this remark. We completely agree that the changing role of Transcatheter Aortic Valve Implantation (TAVI) is an important consideration in the context of aortic valve replacement procedures. In response, we have added a section in the Discussion to address the potential transition from SAVR to TAVI. As the criteria for TAVI eligibility expand, especially for older and higher-risk patients, it is reasonable to predict that some patients who would have previously undergone SAVR may now be considered for TAVI. We have referenced this shift in the manuscript and discussed how the increasing availability and adoption of TAVI may influence future SAVR demand projections, as TAVI could reduce the need for SAVR procedures. However, it is important to note that our study used 30 years of data for SAVR procedures, which allowed us to make reliable projections for the future. Unfortunately, similar long-term data for TAVI is not yet available, as TAVI has only become a standard treatment in recent years. As such, we were unable to include TAVI data in our projections. This, however, is part of our future research plans, as we aim to include TAVI data in our models once sufficient long-term data becomes available. We also emphasize the need for further investigation into the impact of TAVI on long-term healthcare planning and resource allocation for aortic valve replacement therapies. These changes have been highlighted in yellow in the manuscript for your reference and were revised along with those mentioned in comment 2 (lines 357-364).
|
|
Comments 8: I would try to make less local and broader reasoning |
|
Response 8: Your suggestion is appreciated. As you can see, we concurred that an expansion of the scope of the discussion is sure to sharpen the relevance of the outcomes. We responded by rewriting the Discussion section to incorporate greater global insight into the challenge of population aging and the heightened demand for replacement of the aortic valve. We now discuss how the same demographic trends are impacting the healthcare system on the global level, with an emphasis on the developed nations, and the implication of such trends for the planning of care and the allocation of resources in these nations. We provide international studies and data to emphasize how the results of the present study are sure to be extrapolatable to other care settings beyond the local level. These changes have been highlighted in yellow in the manuscript for your reference and were changed together with those in comment 2 (lines 333-343).
|
|
Comments 9: A brief discussion on European needs (and not only referring to your hospital) |
|
Response 9: Thank you for your suggestion. I believe this matter has already been covered in the earlier section, where we extended the discussion to encompass wider European trends concerning population aging, the healthcare burden, and the importance of strategic planning and resource distribution across the continent. Moreover, the analysis of technological advancements and the adoption of eHealth further emphasizes the broader European context, in line with the necessity for Europe-wide approaches to tackle the issues associated with an aging population. Consequently, I believe the discussion effectively addresses the European requirements beyond the scope of our hospital.
|
|
Comments 10: Can we foresee a transition of these patients to private healthcare? |
|
Response 10: I appreciate you bringing about this key consideration. It is certainly the case that future patients who need SAVR may end up receiving treatment in the private sector, particularly as demand for cardiovascular procedures is set to rise as populations grow older. Although public healthcare systems can expect to feel the strain of demand, particularly in areas where resources are scarce, the private sector is perhaps better poised to meet the demand. This may be the trend in countries with greater access to private healthcare as well as where there is the capability to pay for private treatment. We have discussed the potential in the Discussion section, referring to the necessity for overall planning that considers public and private health resources in the management of demand for aortic valve replacement therapies. These changes have been highlighted in yellow in the manuscript for your reference and were changed together with those in comment 2 (lines 357-359).
|
|
4. Response to Comments on the Quality of English Language |
|
“The English is fine and does not require any improvement.” |
|
No comments needs.
|
|
5. Additional clarifications |
|
Nothing to add |
Reviewer 2 Report
Comments and Suggestions for Authors
The authors of the manuscript present results from an original study, aiming to predict the increase of number of patients indicated for surgical aortic valve replacement (SAVR) at a certain region of Portugal. The study is important from clinical, social and health system financial point of view. Degenerative aortic stenosis is the most common valvular heart disease requiring interventional treatment in most countries worldwide. The prevalence of this condition is expected to rise additionally in the next decades due to population aging in many parts of the world. Surgical replacement continues to be first-line treatment for patients with severe Ao stenosis who are symptomatic or have additional criteria in the lack of symptoms, particularly patients aged <75 years with non-elevated surgical risk (STS Prom/Euroscore II <4%).
The manuscript is generally well-structured. The text is plain, the tables - informative and the figures - illustrative. Yet, I have some recommendations to the authors, which I belive will improve the scientific quality of their work:
1. The title must be improved - it does not sound quite well in its current version.
2. Two sentences in the Introduction are missing citation indexes (lines 45-47 and 58-60).
3. Clear inclusion/exclusion criteria for patient enrollment in this study must be formulated.
4. If possible, I recommend to the authors to make an additional analysis (if not possible for this article - mandatory for the next one on this topic) comparing the predicted increase of the number of patients who will be treated by SAVR to patients who will be referred for transcatheter aortic valve replacement (TAVR).
5. English language could be improved at some parts of the text.
Author Response
|
Comments 1: The title must be improved - it does not sound quite well in its current version. |
|
Response 1: Thank you for pointing this out. We value your input and have taken it into consideration. Following careful thought, we have changed the title to one which is brief and effective. The revised title is:
"Predicting the Demand for Surgical Aortic Valve Replacement in a Tertiary Centre: The Aged Population for the Next Decades"
We hope this revised title better reflects the content of the study and improves its clarity and appeal. Please let us know if you feel this change addresses the issue.
|
|
Comments 2: Two sentences in the Introduction are missing citation indexes (lines 45-47 and 58-60). |
|
Response 2: Thank you for pointing this out. We have reviewed the sentences in question and added the appropriate citation indexes to ensure proper referencing. The citations have now been included for the sentences on lines 45-47 and 58-60.
|
|
Comments 3: Clear inclusion/exclusion criteria for patient enrollment in this study must be formulated. |
|
Response 3: Thank you for this valuable suggestion. We agree that clear inclusion and exclusion criteria are essential for ensuring the robustness and reproducibility of the study. We have updated the Methods section to reflect these criteria, ensuring transparency and clarity in the patient selection process. These changes have been highlighted in yellow in the manuscript for your reference (lines 119-121).
|
|
Comments 4: If possible, I recommend to the authors to make an additional analysis (if not possible for this article - mandatory for the next one on this topic) comparing the predicted increase of the number of patients who will be treated by SAVR to patients who will be referred for transcatheter aortic valve replacement (TAVR). |
|
Response 4: Thank you for this remark. We completely agree that the changing role of Transcatheter Aortic Valve Implantation (TAVI) is a consideration to be taken into account in the setting of procedures for the replacement of the aortic valve. In response, we introduced an additional section within the Discussion to provide for the potential transition from SAVR to TAVI. However, we are not able to include this data in this work at this stage. As the criteria for TAVI eligibility broaden further, especially in the setting of older and higher-risk patients, it is reasonable to predict that certain patients who previously underwent SAVR might now be candidates for TAVI. We have referenced the transition in the manuscript and described how the trend might impact future demand projections for SAVR as the enhanced availability and use of TAVI may diminish the demand for SAVR procedures. We emphasize as well the need for further investigation into the effect of TAVI on long-term health care planning and the allocation of resources for therapies in the replacement of the aortic valve. In the same manner that we include a potential future analysis to this paradigm in TAVRS set. These changes have been highlighted in yellow in the manuscript for your reference (lines 357-364).
|
|
Comments 5: English language could be improved at some parts of the text. |
|
Response 5: Thank you for this remark We have reviewed the manuscript thoroughly and made revisions to enhance the fluency and the clarity of the language. We have endeavoured to make the text as concise and as precise as possible, and to strengthen the areas where the language may be less clear or less polished. We hope these revisions do contribute to improving the readability of the manuscript. If further improvements are desired, let us know.
|
|
4. Response to Comments on the Quality of English Language |
|
“The English could be improved to more clearly express the research.” |
|
Responded in #5 comment.
|
|
5. Additional clarifications |
|
Nothing to add |
Round 2
Reviewer 2 Report
Comments and Suggestions for Authors
The authors have observed most of the reviewer's recommendations and have made the prosposed corrections/additions to their text.
The manuscript scientific quality has been subtsantially improved and now it can be considered by the Edtiors of JCM for publication in its current version.